# Improving Sample-based Evaluation for Generative Adversarial Networks

## Abstract

In this paper, we address quantitative sample-based evaluation for Generative Adversarial Networks (GANs) on generating domain-specific images, where we improve conventional sample-based evaluation methods on two levels: the feature representation and the evaluation metric. Unlike most existing evaluation frameworks which transfer the representation of ImageNet inception model for mapping images onto the feature space, our method uses a specialized encoder to acquire fine-grained domain-specific representation. Moreover, for datasets with multiple classes, we propose Class-Aware Frechet Distance (CAFD), which employs a Gaussian mixture model on the feature space to better fit the feature distribution with multiple clusters. Experiments and analysis on both the feature level and the image level were conducted to demonstrate improvements of our proposed framework over the recently proposed state-of-the-art sample-based method FID. To our best knowledge, we are the first to provide counter examples where FID gives inconsistent results with human judgments. It is shown in the experiments that our framework is able to overcome the shortness of FID and improves robustness. Code will be made available.

## 1 Introduction

Generative Adversarial Networks (GANs) have shown outstanding abilities on many computer vision tasks including generating domain-specific images (Goodfellow et al., 2014a), style transfer (Zhu et al., 2017), super resolution (Ledig et al., 2017), etc. The basic idea of GANs is to hold a two-player game between generator and discriminator, where the discriminator aims to distinguish between real and fake samples while the generator tries to generate samples as real as possible to fool the discriminator.

Researchers (Radford et al., 2016; Arjovsky et al., 2017; Berthelot et al., 2017; Mao et al., 2017) have been continuously exploring better GAN architectures. However, developing a widely-accepted GAN evaluation framework remains to be a challenging topic (Theis et al., 2016). Due to a lack of GAN benchmark results, newly proposed GAN variants are validated on different evaluation frameworks and therefore incomparable. Because human judgements are inherently limited by manpower resource, good quantitative evaluation frameworks are of very high importance to guide future research on designing, selecting, and interpreting GAN models.

There have been varieties of efforts on designing sample-based evaluation for GANs on its ability of generating domain-specific images. The goal is to measure the distance between the generated samples and the real in the dataset. Most existing methods utilized the ImageNet (Russakovsky et al., 2015) inception model to map images onto the feature space. The most widely used criteria is probably the Inception Score (Salimans et al., 2016), which measures the distance via Kullback-Leiber Divergence (KLD). However, it is probability based and is unable to report overfitting. Recently, Frechet Inception Distance (FID) was proposed (Heusel et al., 2017) on improving Inception Score. It directly measures Frechet Distance on the feature space with the Gaussian assumption. It has been proved that FID is far better than Inception Score (Huang et al., 2018; Im et al., 2018; Lucic et al., 2017). However, we argue that assuming normality on the whole feature distribution may lose class information on labeled datasets.

In this work, we propose an improved quantitative sample-based evaluating criteria. We improve conventional evaluation methods on two levels: the feature representation and the evaluation metric.

Unlike most existing methods including the Inception Score (Salimans et al., 2016) and FID (Heusel et al., 2017), our framework uses a specialized encoder trained on the dataset to get domain-specific representation. We argue that applying the ImageNet model to either labeled or unlabeled datasets is ineffective. Moreover, we propose Class-Aware Frechet Distance (CAFD) in our framework to measure the distribution distance of each class (mode) respectively on the feature space to include class information. Instead of the single Gaussian assumption, we employ a Gaussian mixture model (GMM) to better fit the feature distribution. We also include KL divergence (KLD) between mode distribution of real data and generated samples into the framework to help detect mode dropping.

Experiments and analysis on both the feature level and the image level were conducted to demonstrate the improved effectiveness of our proposed framework. To our best knowledge, we are the first (Borji, 2018) to provide counter examples where FID is inconsistent with human judgements (See Figs. 1 and 2). It is shown in the experiments that our framework is able to overcome the shortness of existing methods.

## 2 RELATED WORK

**Evaluation Methods.** Several GAN evaluation methods have been proposed by researchers. While model-based methods including Parzen window estimation and the annealed importance sampling (AIS) (Wu et al., 2017) require either density estimation or observation on the inner structure of the decoder, model-agnostic methods (Heusel et al., 2017; Salimans et al., 2016; Im et al., 2018; Che et al., 2017; Dziugaite et al., 2015; Lopez-Paz & Oquab, 2017; Li et al., 2015) are more popular in the GAN community. These methods are sample based. Most of them map images onto the feature space via an ImageNet pretrained model and measure the similarity of the distribution between the dataset and the generated data. Maximum mean discrepancy (MMD) was proposed by (Dziugaite et al., 2015; Li et al., 2015) and it has been further used in classifier two-sample tests (Lopez-Paz & Oquab, 2017), where statistical hypothesis testing is used to assess whether two sample sets are from the same distribution. Inception Score (Salimans et al., 2016), along with its improved version Mode Score (Che et al., 2017), was the most widely used metric in the last two years. FID (Heusel et al., 2017) was proposed on improving the Inception Score. Recently, several interesting methods were also proposed including classification accuracy (Shmelkov et al., 2018), precision-recall measuring (Sajjadi et al., 2018) and skill rating (Olsson et al., 2018). These metrics give complementary perspectives towards sample-based methods.

**Studies on Existing Frameworks.** It is common (Barratt & Sharma, 2018) in the literature to see algorithms which use existing metrics to optimize early stopping, hyperparameter tuning, and even model architecture. Thus, comparison and analysis on previous evaluation methods have been attracting more and more attention recently (Theis et al., 2016; Huang et al., 2018; Im et al., 2018; Lucic et al., 2017). While Inception Score was the most popular metric in the last two years, it was believed to be misleading in recent literature (Heusel et al., 2017; Huang et al., 2018; Lucic et al., 2017; Borji, 2018; Barratt & Sharma, 2018). Applying the ImageNet model to encode features in Inception Score is ineffective (Theis et al., 2016; Barratt & Sharma, 2018; Rosca et al., 2017). The recently proposed FID has been proved to be far better than Inception Score (Heusel et al., 2017; Huang et al., 2018; Im et al., 2018). And its robustness was experimentally demonstrated recently in a technical report (Lucic et al., 2017). However, we argue that FID still has problems and provide counter examples where FID gives inconsistent results with human judgements. Moreover, we propose an improved version of evaluation which overcomes its shortness.

## 3 PROBLEMS ON FID

The evaluation problem can be formulated as modeling the distance between two distributions $P_r$ and $P_g$, where $P_r$ denotes the distribution of real samples in the dataset and $P_g$ denotes the distributions of new samples generated by GAN models.

The main difficulties for GANs on generating domain-specific images can be summarized into three types below.

- **Lack of generating ability.** Either the generator cannot generate useful samples or the GAN training cannot diverge.

- **Mode collapse.** Different modes collapse to a new mixed mode in the generated samples. (e.g. An animal resembling both a horse and a deer.)

- **Mode dropping**. Only part of the modes in the dataset are generated while some modes are implicitly ignored. (e.g. The handwritten 5 can hardly be generated by GAN trained on MNIST.)

Therefore, a good evaluation framework should be consistent to human judgements, penalize on mode collapse and mode dropping. Most of the conventional methods utilized an ImageNet pre-trained inception model to map images onto the feature space. Inception Score, which was originally formulated as Eq. (1), ignored information in the dataset completely. Thus, its original formulation was considered to be relatively misleading.

$$IS = exp(E_x[KL(p(y|x)||p(y))]) \tag{1}$$

The Mode Score was proposed (Che et al., 2017) to overcome this shortness. Its formulation is shown in Eq. (2). By including the prior distribution of the ground truth labels, Mode Score improved Inception Score (Che et al., 2017) on reporting mode dropping.

$$MS = exp(E_x[KL(p(y|x)||p(y^*))] - KL(p(y^*)||p(y))) \tag{2}$$

FID (Heusel et al., 2017), which was formulated in Eq. (3), was proposed on improving Inception Score (Salimans et al., 2016).

$$FID(P_r, P_g) = ||\mu_r - \mu_g|| + Tr(C_r + C_g - 2(C_r C_g)^{\frac{1}{2}}) \tag{3}$$

$(\mu_g, C_g)$, $(\mu_r, C_r)$ are the first-order and second-order statistics for generated samples and real data respectively. Unlike the previous two metrics which are probability-based, FID directly measures Frechet distance on the feature space. It uses an ImageNet model for encoding features and assumes normality on the whole feature distribution. FID was believed to be better than Inception Score (Huang et al., 2018; Im et al., 2018; Lucic et al., 2017). However, we argue that FID still has two major problems (See Section 3.1 and 3.2).

### 3.1 INEFFECTIVE REPRESENTATION

As both Inception Score (Salimans et al., 2016) and Mode Score (Che et al., 2017) is probability-based, applying the ImageNet pretrained model on non-ImageNet dataset is relatively meaningless. This misuse of representation on Inception Score was mentioned previously (Rosca et al., 2017). However, we argue that applying the ImageNet model to map the generated images to the feature space in FID can also be misleading.

While both of the (Barratt & Sharma, 2018; Rosca et al., 2017) mentioned that applying the ImageNet pretrained model to the probability-based metric Inception Score (Salimans et al., 2016) is inadequate, the trend for applying it to feature-based metric such as FID (Heusel et al., 2017) is widely followed. (Barratt & Sharma, 2018) pointed out that because classes are unmatched, the $p(y|x)$ and $p(y^*)$ in the formulation of Inception Score are meaningless. However, we argue that applying the ImageNet model for mapping the generated images to the feature space in FID can also be misleading for the two reasons below.

First, On labeled datasets with multiple classes, the class labels unmatch those in ImageNet. For example, the class 'Bird' in CIFAR-10 (Krizhevsky & Hinton, 2009) is divided into several sophisticated category labels in ImageNet. This will make the CNN representations trained on ImageNet is either meaningless or over-complicated. Specifically, some features distinguishing the "acoustic guitar" from "electric guitar" are hardly useful on CIFAR-10 while fine-grained features distinguishing "African hunting dog" from "Cape hunting dog" (which all belong to the category "dog" in CIFAR-10) are not needed as well.

On unlabeled datasets with images from a single class such as CelebA (Liu et al., 2015), applying the ImageNet inception model is also inappropriate. The categories of ImageNet labels are so sophisticated that the trained model needs to encode diverse features on various objects. However, this will get encoded features limited to a relatively low-dimensional subspace lack of fine-grained information. For example, the ImageNet models can hardly distinguish different faces. In Section 5.1, we designed experiments on both the feature level and the image level to demonstrate the effects of using different representations.

## 3.2 SINGLE GAUSSIAN VS. MIXTURE GAUSSIAN

We argue that the single Gaussian assumption in FID is over-simplified. As the training decreases intra-class distance and increases inter-class distance, the features are distributed in groups by their class labels. Thus, we propose that on datasets with multiple classes, the feature distribution is better fitted by a Gaussian mixture model.

Considering the specific Gaussian mixture model where $x \sim N(\mu_i, C_i)$ with probability $p_i$, we can derive the first and second moment of the feature distribution in Eq. (4) and Eq. (5).

$$\mu = E(x) = E(E(x|y)) = \sum p_i \mu_i \tag{4}$$

$$C = \text{var}(x) = E(\text{var}(x|y)) + \text{var}(E(x|y))$$
$$= \sum p_i C_i + \sum p_i (\mu_i - \mu)(\mu_i - \mu)^T \tag{5}$$

It should be noted that when the feature is n-dimensional and there are $K$ classes in total, there are a total of $K(\frac{n^2+n}{2} + n + 1)$ variables in the model. However, directly modeling the whole distribution Gaussian as in FID will result in $\frac{n^2+n}{2} + n$ degrees of freedom, which is a relatively small number. Thus, FID detects mode-related problems in an implicit way. Either simply dropping a mode or linearly combining images increases FID by unintentionally changing the mean $\mu$. However, FID gets to be misleading when the deficiency type becomes more complicated (See Figure 2).

## 4 PROPOSED FRAMEWORK

### 4.1 DOMAIN-SPECIFIC ENCODER

As discussed in Section 3.1, applying the ImageNet inception model to either labeled or unlabeled datasets is ineffective. We argue that a specialized domain-specific encoder should be used for sample-based evaluation. While the features encoded by the ImageNet model are limited within a low-dimensional subspace, the domain-specific model could encode more fine-grained information, making the encoded features much more effective. Specifically, we propose to use the widely used variational autoencoder (VAE) (Kingma & Welling, 2014) to acquire the specialized embedding for a specific dataset. In labeled datasets, we can add a cross-entropy loss for training the VAE model. In Section 5.1, we show that simply training an autoencoder can already get better domain-specific representations on CelebA (Liu et al., 2015).

### 4.2 CLASS-AWARE FRECHET DISTANCE

Before introducing our improved evaluation metric, we would firstly take a step back towards existing popular metrics. Both Inception Score (Salimans et al., 2016) and Mode Score (Che et al., 2017) measure distance between probability distribution while FID (Heusel et al., 2017) directly measures distance on the feature space. Probability-based metrics better handle mode-related problems (with the correct use of a domain-specific encoder), while directly measuring distance between features better models the generating ability. In fact, we believe these two perspectives are complementary. Thus, we propose a class-aware metric on the feature space to combine the two perspectives together. For datasets with multiple classes, the feature distribution is better fit with mixture Gaussian (See Section 3.2). Thus, we propose Class-Aware Frechet Distance (CAFD) to include class information. Specifically, we compute probability-based Frechet Distance between real data and generated samples in each class respectively.

Table 1: Frechet distance on different classes of MNIST dataset.

| class | 0 | 1 | 2 | 3 | 4 | 5 |
|---|---|---|---|---|---|---|
| dist | $64.8 \pm 0.5$ | $18.9 \pm 0.2$ | $80.5 \pm 1.1$ | $81.3 \pm 0.3$ | $64.5 \pm 0.6$ | $79.0 \pm 0.4$ |
| class | 6 | 7 | 8 | 9 | average | |
| dist | $65.2 \pm 0.3$ | $46.8 \pm 0.3$ | $90.4 \pm 0.3$ | $59.8 \pm 0.2$ | $65.1 \pm 0.4$ | |

As previously discussed in Section 4.1, we train a domain-specific VAE along with the cross entropy on datasets with multiple classes and use its learned representations. In our evaluation framework, we also made use of the predicted probability $p(y|x)$. To calculate the expected mean of each class in a specific set $S$ of generated samples, we can derive the formulation below in Eq. (6).

$$
\begin{aligned}
\mu_i^g = E[x|y_i] &= \sum_{x_j \in S} x_j \mathrm{p}(x_j|y_i) = \sum_{x_j \in S} x_j \frac{\mathrm{p}(x_j, y_i)}{\mathrm{p}(y_i)} \\
&= \sum_{x_j \in S} x_j \frac{\mathrm{p}(y_i|x_j)\mathrm{p}(x_j)}{\sum_{x^* \in S} \mathrm{p}(y_i|x^*)\mathrm{p}(x^*)} \\
&\overset{i.i.d}{=} \sum_{x_j \in S} x_j \frac{\mathrm{p}(y_i|x_j)}{\sum_{x^* \in S} \mathrm{p}(y_i|x^*)} = \sum_{x_j \in S} w_{ij} x_j
\end{aligned}
\tag{6}
$$

where

$$
w_{ij} = \frac{\mathrm{p}(y_i|x_j)}{\sum_{x^* \in S} \mathrm{p}(y_i|x^*)}
\tag{7}
$$

Similarly, The covariance matrix in each class is shown in Eq. (8).

$$
C_i^g = \sum_{x \in S} w_{ij}(x_j - \mu_i)(x_j - \mu_i)^T
\tag{8}
$$

We compute Frechet distance in each of the $K$ classes and average the results to get Class-Aware Frechet Distance (CAFD) in Eq. (9).

$$
CAFD(P_r, P_g) = \frac{1}{K} \sum_{i=1}^{K} \{||\mu_i^r - \mu_i^g|| + Tr(C_i^r + C_i^g - 2(C_i^r C_i^g)^{\frac{1}{2}})\}
\tag{9}
$$

This improved form based on mixture Gaussian assumption can better evaluate the actual distance compared to the original FID. Moreover, when CAFD is applied to evaluating a specific GAN model, we could get better class-aware understanding towards the generating ability. For example, as shown in Table 1, the selected model generates digit 1 well but struggles on other classes. This information will provide guidance for researchers on how well their generative models perform on each mode and may explain what specific problems exist.

As both FID and CAFD aim to model how well domain-specific images are generated, they are not designed to deal with mode dropping, where some of the modes are missed in the generated samples. Thus, motivated by Mode Score (Che et al., 2017), we propose that KL divergence $KL(p(y^*)||p(y))$ should be included as auxiliary scores into the evaluation framework.

To sum up, the correct use of encoder, the CAFD and the KL divergence term combine for a complete sample-based evaluation framework. Our proposed method combines the advantages of Inception Score (Salimans et al., 2016), Mode Score (Che et al., 2017) and FID (Heusel et al., 2017) and overcomes their shortness.

## 4.3 DISCUSSION

Our method is sensitive to different representations. Different selection of encoders can result in changes on the evaluation results. Experiments in Section 5.1 demonstrate that the ImageNet inception model will give misleading results (See Figure 1). Thus, a domain-specific encoder should be used in each evaluation pipeline. Because the representation is not fixed, the correct use (with

Table 2: Results on the explained variance of principle component analysis (PCA) on features encoded by different represenations. 'Exp' denotes 'explained variance', 'Ac-Exp' denotes 'accumulated explained variance'. Although the architecture of ImageNet inception model is much more complex than the domain-specific autoencoder and VAE, the features encoded by ImageNet model are limited in a relatively low-dimensional subspace.

| Component | ImageNet | | Autoencoder | | VAE | |
|---|---|---|---|---|---|---|
| | Exp | Ac-Exp | Exp | Ac-Exp | Exp | Ac-Exp |
| 1 | 9.35% | 9.35% | 5.58% | 5.58% | **3.02%** | **3.02%** |
| 2 | 7.04% | 16.39% | 4.66% | 10.24% | **2.24%** | **5.26%** |
| 3 | 3.88% | 20.27% | 3.93% | 14.17% | **2.08%** | **7.34%** |
| 4 | 2.67% | 22.95% | 3.66% | 17.83% | **2.00%** | **9.34%** |
| 5 | 2.47% | 25.42% | 3.41% | 21.24% | **1.91%** | **11.25%** |

domain-specific representation) of Inception Score, Mode Score and FID all suffer from this sensitivity problem. It is worth emphasizing that different generative methods should be compared only under the same encoder.

Unlike Inception Score, because CAFD measures distance on the feature space as FID does, it is able to report overfitting. By measuring CAFD with respect to training set and test set respectively, researchers can get understanding towards whether their GAN models overfit the training data. Moreover, the intermediate results can provide researchers comprehensive understanding towards their GAN models (e.g. See Table 1).

## 5 EXPERIMENTS

### 5.1 STUDY ON REPRESENTATION

In this section, we study the representation for mapping the generated images onto the feature space. As discussed in Section 4.1, applying the pretrained ImageNet inception model to sample-based evaluation methods is inappropriate. We firstly investigated the features generated by different encoders on CelebA (Liu et al., 2015), which is a widely used dataset containing more than 200k face images. Then, we gave an intuitive demonstration where FID (Heusel et al., 2017) using ImageNet pretrained representations gives inconsistent results with human judgements.

We give two proposals of domain-specific encoders in the experiment: an autoencoder and a VAE (Kingma & Welling, 2014). Both proposed encoders share a similar network architecture which is the inverse structure of the 4-conv DCGAN (Radford et al., 2016). The embedding is dimensioned 2048, which is the same as the dimension of ImageNet features. We train both models for 25 epochs. The loss weight of the KLD term in VAE is 1e-5.

### 5.1.1 FEATURE ANALYSIS

We conducted principle component analysis (PCA) on three feature sets encoded on CelebA (Liu et al., 2015): 1) ImageNet inception model. 2) proposed autoencoder 3) proposed VAE. Table 2 shows the percent of explained variance on the first 5 components. Although the ImageNet model should have much greater representation capability than the 4-conv encoder, its first two components has much higher explained variance (9.35% and 7.04%). This supports our claim that the features encoded by ImageNet are limited in a low-dimensional subspace. It can be also noted that VAE better makes use of the feature space compared to the naive autoencoder.

### 5.1.2 CELEBA: HACK AND COMPARISON

To better demonstrate the deficiency of the ImageNet model, we performed three different types of adjustments on the first 10,000 images on CelebA (Liu et al., 2015): a) Random noise uniformly distributed in [-33,33] was applied on each pixel. b) Each image was divided into 8x8=64 regions and seven of them were sheltered by a pixel sampled from the face. c) Each image was first divided into 4x4=16 regions and random exchanges were performed twice.

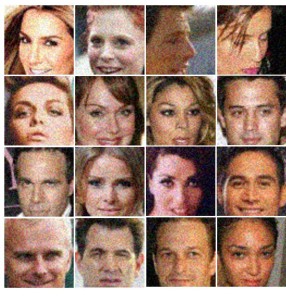 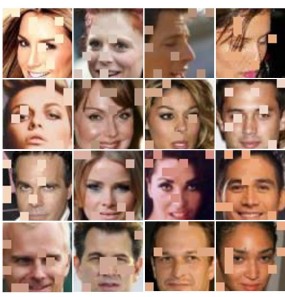 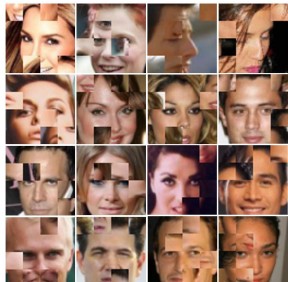

(a) noise (FID=75.9)         (b) sheltering (FID=74.3)         (c) exchange (FID=70.9)

Figure 1: Examples where FID gives inconsistent results with human judgements ($a < b < c$) on CelebA (Liu et al., 2015). The ImageNet inception model fails to encode fine-grained features on faces. a) Random noise uniformly distributed in [-33,33] was applied on each pixel. b) Each image was divided into 8x8=64 regions and seven of them were sheltered by a pixel sampled from the face. c) Each image was first divided into 4x4=16 regions and random exchanges were performed twice.

Table 3: FID results on different representations. Only domain-specific encoders including AE and VAE used in our proposed method provides consistent results with human judgements.

|               | noise    | sheltering | exchange |
|---------------|----------|------------|----------|
| ImageNet      | 76       | 74         | **71**   |
| Discriminator | 122466   | 48322      | **28557**|
| AutoEncoder   | **83**   | 21417      | 38609    |
| VAE           | <**1.0** | 41.1       | 111.8    |
| Human         | **Good** | Bad        | Worst    |

Results are shown in Figure 1. With the ImageNet inception model, it is obvious that FID gave inconsistent results with human judgements (See Table 5). In fact, when similar adjustments were conducted with the overall color maintained, FID fluctuated within only a small range. The ImageNet model mainly extracts general features on color, shape to better classify objects in the world while domain-specific facial textures cannot be well represented.

For comparison, we applied the trained autoencoder and VAE onto the case. Also, we tried to apply the representation of the discriminator after GAN training, which was previously proposed in (Che et al., 2017). Specifically, we use the features right before the final fc layer for the discriminator. Results are shown in Table 3. It is shown that only representations derived from the domain-specific encoder including the autoencoder and VAE are effective and give results consistent with human judgements. The discriminator which learns to discriminate fake samples from the real cannot learn good representation for distance measurement.

Thus, for datasets where images are from a single class such as CelebA (Liu et al., 2015) and LSUN Bedrooms (Yu et al., 2015), the representation should be acquired via training a domain-specific encoder such as a VAE. In this way our sample-based evaluation employs specialized representations, which can provide more fine-grained information related to the specific domain.

## 5.2 STUDY ON EVALUATION METRIC

In this section, we used the domain-specific representations and studied the improvements of the evaluation metric CAFD proposed in our framework against the state-of-the-art metric FID (Heusel et al., 2017). In datasets with multiple classes, the Gaussian mixture model in CAFD will better fit the feature distribution. First, we performed user study to demonstrate the improved consistency of our method. Then, An intuitive case for further demonstration is given where CAFD shows great robustness while FID fails to give consistent results with human judgements. For implementation details, on the MNIST dataset, we trained a variational autoencoder (VAE) (Kingma & Welling, 2014) with the kl loss weight 1e-5 for the specialized encoder and added the cross-entropy term with a loss weight of 1.0.

Table 4: The fraction of pairs on which each metric agrees with human judgements on MNIST. The count of agreed pairs over the count of total pairs (210) is reported.

| Method | easy | hard |
|---|---|---|
| Inception Score | 158 / 210 | 104 / 210 |
| Mode Score | 172 / 210 | 118 / 210 |
| FID | 201 / 210 | 142 / 210 |
| Ours | **206** / 210 | **151** / 210 |

Table 5: Human judgements on Figure 1.

| Vote for the best | (a) | (b) | (c) |
|---|---|---|---|
| 1st | **23** | 2 | 0 |
| 2nd | 2 | **23** | 0 |
| 3rd | 0 | 0 | **25** |

Table 6: Human judgements on Figure 2.

| | (a) | (b) |
|---|---|---|
| Vote for the better | **25** | 0 |

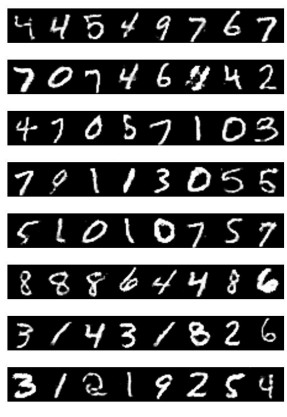

(a) generated (FID=49.9)          (b) hack (FID=25.4)

Figure 2: Examples where FID gives inconsistent results with human judgements on MNIST. Due to the over-simplified Gaussian assumption, FID can be hacked by mode collapse. a) Samples generated by a GAN model. b) Handmade images via axis permutation and FGSM (Goodfellow et al., 2014b). We use the domain-specific representation of VAE for embedding images. (See Table 7)

### 5.2.1 USER STUDY

Evaluating the evaluation metrics is a non-trivial task, as the best criterion is the consistency with human judgements. Therefore, we performed user study to compare our proposed method with the existing ones including Inception Score (Salimans et al., 2016), Mode Score (Che et al., 2017) and FID (Heusel et al., 2017). Our setting is consistent with (Im et al., 2018). 15 volunteers were first trained to tell generated samples from the groundtruth in the dataset. Then, paired image sets were randomly sampled and volunteers were asked to tell the better sets. Finally, we counted pairs where the metric agreed the voted results by the volunteers. We conducted experiments on MNIST with two settings for the experiments: 'easy' and 'hard'. The 'easy' setting is where random pairs are sampled from the intermediate results of GAN training, while the 'hard' setting is where only random pairs with the difference of FID of two sampled sets within a threshold are included. Table 4 shows the results. It is worth noting that in hard cases, the results of Inception Score (Salimans et al., 2016) are relatively meaningless (50%), which makes it hard to be applied as guidance for improving the quality of generated images by GANs. In both 'easy' and 'hard' settings, our method gets consistent gain compared to baseline approaches.

### 5.2.2 MNIST: HACK AND COMPARISON

In this experiment, we gave an intuitive case where FID fails to give consistent results with human judgements. We used two different settings of representations and focused on the evaluation metric within each setting. Specifically, Besides the VAE, we also train a classifier on MNIST and use its representation as a supporting experimental setting.

Table 7: Results of FID, CAFD and KLD on MNIST. Lower scores infer better image quality. The 'test' denotes the MNIST test set, 'adjusted' denotes the features after axis permutation. 'generated' denotes samples generated by a specific GAN model. 'hack' denotes the image sets after FGSM (Goodfellow et al., 2014a). We use two setting of different representations in this experiment: a domain-specific classifier and a VAE. For VAE, 'generated' and 'hack' are the sampled images in Figure 2. Compared to FID, CAFD are more robust to feature-level adjustments.

| classifier | FID | CAFD | VAE | FID | CAFD |
|---|---|---|---|---|---|
| test | 0 | 0 | test | 0 | 0 |
| adjusted | 0 | 539.8 | adjusted | 0 | 246.2 |
| generated | 73.1 | **201.4** | generated | 49.9 | **80.7** |
| hack | 72.8 | **468.6** | hack | 25.4 | **211.6** |
| train | 22.0 | 99.8 | train | 6.0 | 31.5 |

FID, as an overall statistical measure, is able to detect either a single mode dropping or a trivial linear combination of two images. However, as its formulation has relatively limited constraints, it can be hacked in complicated scenarios. Considering the features extracted from MNIST test data, which has a zero FID with itself. We performed operations below on the features.

Step 1  Performed principle component analysis (PCA) on the original features.

Step 2  Normalized each axis to zero mean and unit variance.

Step 3  Switched the normalized projection of the first two component.

Step 4  Unnormalized the data and reconstructed features.

The adjusted features are completely different with the original one with zero FID maintained. The over-simplified Gaussian assumption on overall distribution cannot tell the differences while our proposed method is able to report the changes with CAFD raising from 0 to 246.2 (539.8) for VAE (classifier). (See Table 7)

Furthermore, We used FGSM (Goodfellow et al., 2014b) to reconstruct the images from the adjusted features in both settings. Specifically, we first trained an decoder for initialization via an AutoEncoder with the encoder fixed. Then, we performed pixelwise adjustment via FGSM (Goodfellow et al., 2014b) to lower the reconstruction error. Because the used encoder has a relatively simple structure, the final reconstruction error is still relatively high after optimized. For comparison, We trained a simple WGAN-GP (Gulrajani et al., 2017) model and took samples (generated by intermediate models during training) with comparable FID with our constructed images. Visualization for the VAE setting are shown in Figure 2.

It is obvious that the quality of constructed images are much worse than the generated samples. After axis permutation, the constructed images suffers from mode collapse. There are many pictures in the right which resemble more than one digits and are hard to recognize. However, for the VAE (classifier) setting, it still received a FID of 25.4 (72.8) lower than 49.9 (73.1) received by generated samples. For comparison, The results of CAFD on these cases are shown in Table 7. While FID gives misleading results, CAFD are much more robust on the adjusted features. Compared to the constructed images (211.6 (468.6)), the generated images received a much lower CAFD (80.7 (201.4)), which is consistent with human judgements. (See Table 6) Thus, results for both settings demonstrates the improved effectiveness of the evaluation metric in our proposed evaluation framework.

## 6  CONCLUSIONS

In this paper, we aimed to tackle the very important problem of evaluating the Generative Adversarial Networks. We presented an improved sample-based evaluation, which improves conventional methods on both representation and evaluation metric. We argue that a domain-specific encoder is needed and propose Class-Aware Frechet Distance to better fit the feature distribution. To our best knowledge, we are the first to provide counter examples where the state-of-the-art FID method is inconsistent with human judgements. Experiments and analysis on both the feature level and the image level have shown that our framework is more effective.

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

Table 8: The classification results on CIFAR-10 (Krizhevsky & Hinton, 2009) images using inception model trained on ImageNet. The class labels 'Bird' and 'Dog' are divided into several subclasses.

| Rank | CIFAR-10 'Bird' | Frequency | CIFAR-10 'Dog' | Frequency |
|---|---|---|---|---|
| 1 | Fox Squirrel | 10.1% | Japanese spaniel | 9.8% |
| 2 | Limpkin | 6.9% | Dandie Dinmont | 5.2% |
| 3 | Black Stork | 6.4% | English foxhound | 4.6% |
| 4 | Black Grouse | 5.3% | Toy terrier | 3.2% |
| 5 | Brambling | 4.1% | Bluetick | 2.8% |

APPENDIX

## A  MISMATCHED IMAGENET LABELS

We used Inception-v3 (Szegedy et al., 2015) model trained on ImageNet to classify the 5000 images labeled 'Bird' and 5000 images labeled 'Dog' in CIFAR-10 (Krizhevsky & Hinton, 2009) dataset respectively. Table 8 shows the results. The images from the single class 'Bird' in CIFAR-10 is classified into various subclasses, where surprisingly the top class is Fox Squirrel (which is not a Bird class) with a 10.1% frequency. The classification results are extremely diverse. It can be inferred that the Inception-v3 model trained on ImageNet does not map images with the label 'Bird' onto a simple subspace. Results on the label 'Dog' show similar patterns. Features determining whether a dog is a Japanese spaniel or an English foxhound are relatively unnecessary on CIFAR-10. Thus, the ImageNet representation cannot well fit non-ImageNet datasets.

Therefore, the encoder should be specifically trained for datasets of which the labels are different from ImageNet. To attain effective representations on non-ImageNet datasets, we need to ensure that the class labels of data used for training GAN models are consistent with those of data used for training the encoder.

## B  NORMALITY TEST

The Gaussian assumption on the features were commonly used in the literature. Although there are non-linear operations such as relu and max-pooling in the neural network, assuming the normality simplifies the model and enables numerical expression. However, in labeled dataset with multiple classes, the Gaussian assumption is relatively over-simplified.

In this experiment, we performed Anderson-Darling test (AD-test) (Scholz & Stephens, 1987) to quantatively study the normality of the data. Specifically, to test the multivariate normality on a set of features, we first performed principle component analysis (PCA) on the data, and then applied AD-test to the first 10 components and averaged the results. We compared the test results on each class and the whole training set on MNIST. We used a simple 2-conv structure trained on the MNIST classification task as our feature encoder with a output dimension 1024. To reduce the influence of sample number on the results, we divided the whole features randomly into 10 sets to study the normality of the mixed features. Results are shown in Table 9. Although the p-value of both features are small, features within a single class get much greater results than the mixed features. It can be inferred that compared to the whole training set, features within each class are much more Gaussian. Thus, the basic assumption of CAFD in our proposed framework is more reasonable compared to the FID (Heusel et al., 2017) method.

## C  A BENCHMARK FOR POPULAR GANS

The idea of Generative Adversarial Network was originally proposed in (Goodfellow et al., 2014a). It has been applied to various computer vision tasks (Zhu et al., 2017; Ledig et al., 2017; Zhu et al., 2016; Isola et al., 2017). Researchers have been continuously developing better GAN architectures (Gurumurthy et al., 2017; Huang et al., 2017) and training strategies (Arora et al., 2017; Hoang et al., 2018) on generating domain-specific images. Deep convolutional networks were firstly in-

Table 9: P-value results of AD-test (Scholz & Stephens, 1987) on features of each class and the whole training images. The whole features were randomly divided into 10 sets. Compared to the mixed features, features encoding images from a single class are more Gaussian.

| set number | 0 | 1 | 2 | 3 | 4 |
|---|---|---|---|---|---|
| class | $1.1 \times 10^{-1}$ | $3.9 \times 10^{-19}$ | $3.8 \times 10^{-3}$ | $7.4 \times 10^{-2}$ | $5.7 \times 10^{-2}$ |
| mixed | $3.2 \times 10^{-13}$ | $1.1 \times 10^{-5}$ | $5.4 \times 10^{-7}$ | $2.2 \times 10^{-11}$ | $9.0 \times 10^{-9}$ |
| set number | 5 | 6 | 7 | 8 | 9 |
| class | $5.6 \times 10^{-2}$ | $9.3 \times 10^{-2}$ | $1.0 \times 10^{-2}$ | $5.0 \times 10^{-2}$ | $2.4 \times 10^{-2}$ |
| mixed | $6.9 \times 10^{-4}$ | $1.4 \times 10^{-12}$ | $2.7 \times 10^{-6}$ | $3.2 \times 10^{-8}$ | $1.7 \times 10^{-11}$ |

Table 10: CAFD Results of different GAN models on MNIST and FASHION-MNIST (Xiao et al., 2017). We use VAE (Kingma & Welling, 2014) trained on specific datasets as the feature encoder.

| | MNIST | FASHION-MNIST (Xiao et al., 2017) |
|---|---|---|
| DCGAN (Radford et al., 2016) | $81.7 \pm 0.6$ | $103.9 \pm 0.7$ |
| LSGAN (Mao et al., 2017) | $75.0 \pm 0.7$ | $55.4 \pm 0.6$ |
| BEGAN (Berthelot et al., 2017) | $72.9 \pm 0.4$ | $69.6 \pm 0.7$ |
| EBGAN (Zhao et al., 2017) | $80.1 \pm 0.4$ | $92.0 \pm 0.8$ |
| DRAGAN (Kodali et al., 2017) | **$64.9 \pm 0.6$** | **$50.9 \pm 0.3$** |
| WGAN (Arjovsky et al., 2017) | $85.5 \pm 0.4$ | $55.9 \pm 0.4$ |
| WGAN-GP (Gulrajani et al., 2017) | **$69.3 \pm 0.4$** | **$48.6 \pm 0.3$** |

troduced to the GAN community by (Radford et al., 2016). Wasserstein GAN (WGAN) (Arjovsky et al., 2017) was proposed to significantly improve convergence on GAN training. Recently, several variants were proposed (Berthelot et al., 2017; Mao et al., 2017; Che et al., 2017; Dziugaite et al., 2015; Zhao et al., 2017; Kodali et al., 2017; Gulrajani et al., 2017) to improve the image quality generated by GAN models.

In order to benmark the performance of GANs on generating domain-specific images, we conducted experiments on 7 popular GAN models[1] including DCGAN (Radford et al., 2016), LSGAN (Mao et al., 2017), BEGAN (Berthelot et al., 2017), EBGAN (Zhao et al., 2017), DRAGAN (Kodali et al., 2017), WGAN (Arjovsky et al., 2017), WGAN-GP (Gulrajani et al., 2017). Our experiments were performed on MNIST and FASHION-MNIST (Xiao et al., 2017). We will include other popular datasets such as CIFAR-10 (Krizhevsky & Hinton, 2009), CelebA (Liu et al., 2015) and ImageNet (Russakovsky et al., 2015) in the future.

Results are shown in Table 10. All of the tested models converge well. DCGAN (Radford et al., 2016), which is the first to introduce convolutional neural networks into generative models, struggles more on convergence than the newly proposed GAN variants. DRAGAN (Kodali et al., 2017) and WGAN-GP (Gulrajani et al., 2017) get the top two scores on both datasets. Both BEGAN (Berthelot et al., 2017) and WGAN (Arjovsky et al., 2017) focus more on stable training, while the qualities of their generated images are not the best. WGAN-GP (Gulrajani et al., 2017) improves WGAN (Arjovsky et al., 2017) by using norm penalizing to replace weight clipping. It generates higher quality images compared to its baseline. DRAGAN (Kodali et al., 2017) utilizes a gradient penalty scheme and mitigates the problem of mode collapse. It is worth noting that the recently proposed DRAGAN (Kodali et al., 2017) and WGAN-GP (Gulrajani et al., 2017) outperform other models by a relatively large margin. We can infer that the development of exploring better GAN architectures and training strategies is still highly active.

---

[1]We used the off-the-shelf tensorflow package `https://github.com/hwalsuklee/tensorflow-generative-model-collections`.

