# OpenReview forum: "Improving Sample-based Evaluation for Generative Adversarial Networks"
_ICLR.cc/2019/Conference_

### Official Review · AnonReviewer2 · 2018-10-28
**Compute a GMM in a learned feature space (AE, VAE) and make it class aware by making use of class information or a prediction thereof.**

**Rating:** 3
**Confidence:** 5

**Review:**

The authors study the task of sample-based quantitative evaluation applied to GANs. The authors suggest multiple modifications to existing evaluation pipelines: (1) Instead of embedding the samples in the InceptionNet feature space, train a domain-specific encoder. If labeled data is available, add a cross-entropy loss to the encoder training objective so that the class can be predicted. (2) Instead of fitting a single Gaussian in the feature space, fit a GMM instead. This should allow for a more fine-grained “class-aware” distance between the (empirical) distributions.

Pro:
Attempt to attack a critical issue in generative modeling. Good overview of competing approaches.
Several ablation studies of evaluation measures and the behavior of FID with respect to the representation space.
The ideas make sense on a conceptual level, albeit suffering from major practical concerns.

Con:
- Clarity can be improved (e.g. use of double negatives as in the top of page 3), the same arguments repeated multiple (>3) times (i.e. deficiencies of FID and IS, etc.), Many statements which should be empirically tested are stated as folklore (last paragraph on page 3). In general the paper merits another polishing pass (mode != model, last paragraph in  section 3, “unmatch”, etc.).
- Why would a VAE capture a good feature space? It is known that the tradeoff between what is stored in the latent space versus the discriminator *completely* depends on the power of the discriminator -- if the discriminator is flexible enough it can just learn the marginal distribution and ignore the latent code. Hence, this subtle issue will likely undermine the entire model comparison.
- Using the predictive distribution as a soft label for CAFD. Interesting idea, but why would one have access to labels in the first place? Why wouldn't one use a conditional GAN if we already have labels? Secondly, why would the modes necessarily correspond to classes?
- Stated issues with FID: Why would you expect FID to be resistant to such drastic transformations as blocking out a significant proportion of pixels with “blocks”? This is a *major* change in the underlying distribution. The fact that humans can “fill in” this gap should have nothing to do with the quality of the underlying model. Arguably, you can also hide one eye, the nose and the mouth and still judge the sample as “good”.

The ideas presented in this paper are conceptually interesting. However, given the drawbacks discussed above I cannot recommend the acceptance of this work.

---

### Official Review · AnonReviewer1 · 2018-10-29
**Interesting ideas but not totally convinced**

**Rating:** 5
**Confidence:** 4

**Review:**

This paper proposes a variant of the popular FID score for evaluating GAN-type generative models. The paper makes two major complaints about the FID as it is currently used:

1. The standard Inception network features trained on ImageNet might not be a good representation for whatever different dataset is being modeled, e.g. CelebA or CIFAR-10.

2. The globally-Gaussian assumption made by the FID doesn't hold, which can cause some problems with the metric.

To address issue 1, the paper proposes choosing features based on a dataset-specific VAE, which can additionally incorporate labels when they're available. For 2, the authors propose to compute something like the FID between each component of a Gaussian mixture, based on soft assignments of points to a class with the VAE's inference network to estimate p(y|x), when labels y are available.

In terms of the definition of the CAFD: it is worth emphasizing that (9) is *not* the Frechet = Wasserstein-2 distance between Gaussian mixtures (which is fine). Rather, it's essentially the mean FID of the class-conditional distributions. This has previously been considered in the conditional GAN case, e.g. by Miyato and Koyama (ICLR 2018, https://openreview.net/forum?id=ByS1VpgRZ ). The difference here is that soft-assignments are supported, through the VAE's inference network, though using any classifier would be essentially equivalent. As long as you have a classifier, you can compute the CAFD, regardless of using a VAE representation or not; the VAE just conveniently gives you a classifier out too. Thus the two components of your proposal are essentially orthogonal.



On the choice of dataset-dependent features:

You say several times through the paper that ImageNet-based features are "ineffective" because the class labels do not match with the target, e.g. "fine-grained features distinguishing 'African hunting dog' from 'Cape hunting dog' (which all belong to the category 'dog' in CIFAR-10) are not needed." This is, I think, somewhat misguided: imagine I took ImageNet and assigned higher-level labels to it, such that each image is only assigned a label at the level "dog," and then trained a GAN on it. Then a classifier wouldn't need to distinguish "African hunting dog" from "Cape hunting dog." But a GAN, which doesn't see the labels at all, is being given *exactly the same problem*, and so the GAN still needs to be able to produce both African hunting dogs and Cape hunting dogs (though it doesn't need to be able to tell the two apart).

Moreover, some people believe that CNNs trained on general-purpose approximate the human visual system reasonably well (for an overview of the arguments, see https://neurdiness.wordpress.com/2018/05/17/deep-convolutional-neural-networks-as-models-of-the-visual-system-qa/ ), and although the overall goals of GANs are somewhat fuzzy, "the distribution appears the same to the human visual system" seems pretty good as a goal.

- I think it's obvious that ImageNet-trained Inception features do not model the human visual system very well on, say, MNIST.

- They're probably also not amazing on CelebA, because it hasn't been fine-tuned for faces the way the human visual system has. (Incidentally, you say that "the ImageNet models can hardly distinguish different faces" -- this needs either a citation or some experimental support, in the appendix, because this is not a well-known fact and seems quite relevant to the common practice of applying ImageNet-trained features to CelebA evaluation.)

- But it's not clear to me that they don't model the human visual system reasonably well on CIFAR-10, or at least a theoretical higher-resolution version of it. It's true that ImageNet models will contain some features specific to distinguishing different types of guitars, and there are no guitars in CIFAR-10. But as long as those features aren't strongly activated by actual images from your model, they shouldn't mess up the distributions you're comparing too much.

So if you're going to argue that ImageNet representations are insufficient on vaguely ImageNet-like tasks such as CIFAR-10, I don't think the arguments you have here are quite convincing. Probably, you need some evidence that the scores are made noisier by the irrelevant features and thus harder to estimate, or else maybe strong empirical evidence that using comparable features specific to the dataset distribution performs better.

Anyway, for datasets that are not very much like ImageNet, using dataset-specific features is clearly sensible and perhaps necessary. But:

- You only provide pretty limited evidence that the VAE is better than a plain autoencoder, namely Table 2 which shows that the VAE puts less information in the top few principal components. But you only show that up to the top 5 components, and in any case it's not obvious that a more-spread distribution would be better.

- An important question that's not really considered here: how much does the FID/CAFD then just measure how well the generative model matches the VAE you get features from? Is it the case that this VAE would give a (nearly-)perfect score under the CAFD, or not?

- The results of Figure 1/Table 3 are very interesting. But I wonder how much of this difference in behavior is due to training on CelebA vs ImageNet and how much is due to the architecture or objective of the autoencoder. It might be interesting to compare to features from an ImageNet VAE and/or a CelebA classifier and see what those say. (The discriminator features are something like a CelebA classifier, but there's other things going on there too.)



On the CAFD versus FID:

Your main argument for the CAFD over the FID is that it is based on a richer model of the distribution, which you claim to be closer to true: the FID is based on a multivariate Gaussian assumption with a total of n + n (n-1)/2 parameters, while you use K times as many parameters. Your Table 9 also gives some slight evidence that the Gaussian mixture gives a better fit to the data than a single Gaussian.

I'm not entirely convinced by Table 9; comparing p-values is in general not necessarily very meaningful, and in particular it seems quite possible that the Anderson-Darling test simply prefers the mixture because the samples are more closely "clumped together" by the VAE than a random subset of inputs. Moreover, in either case the Gaussian assumption is clearly false a priori: in the Inception case, features are the output of a ReLU activation function and hence zero-inflated, and this or something like it may also be the case in your VAE. So comparing the p-value of tests for hypotheses known a priori to be false is probably a misguided endeavor.

But in any case the FID doesn't *really* assume Gaussianity. It coincides with the Frechet / Wasserstein-2 distance between Gaussians, but it's a perfectly plausible semimetric between any pair of distributions that have means and variances. The claim for superiority of CAFD over FID would then need to be something like "the class-conditional means and variances are more representative of the distribution than the global means and variances."

Re: your claim that "As both FID and CAFD aim to model how well domain-specific images are generated, they are not designed to deal with mode dropping" -- this is something of a strange claim, as dropping an entire mode will hopefully affect both the feature mean and especially the variance unless it is done extremely carefully. A related problem, though, is that the CAFD is essentially insensitive to drastically *reweighting* modes, e.g. producing twice as many 1s as 2s on MNIST: if each mode is modeled correctly, the CAFD will not be changed, while the FID would be strongly affected with reasonable features. The Mode Score KL(p(y*) || p(y)) would be sensitive to this, as you suggest, but it feels somewhat hacky.

The type of analysis in Table 1 is interesting, but one issue is that it is sensitive to the scale of each mode in feature space: if your encoder happens to place 1s close together and 2s relatively more spread apart, you'll see a higher conditional FID for 2s than for 1s even if the visual "sample quality" is the same.

One important piece of related work that's missing is Binkowski et al. (ICLR 2018, https://openreview.net/forum?id=r1lUOzWCW ), who demonstrate that the FID estimator is strongly biased in a misleading way. The same problems are inherited by the CAFD, which you should at least mention. Binkowski et al., and independently Xu et al. (https://arxiv.org/abs/1806.07755 ), also proposed using MMD variants on top of Inception features. This has better statistical properties as shown by Binkowski et al., and also explicitly does not make any parametric assumptions about the distribution of features. It would be worth thinking about the relationship of that approach to the FID/CAFD.

Another metric you could compare to is the "Adversarial Divergence" of Yang et al. (ICLR 2017, https://openreview.net/forum?id=HJ1kmv9xx ) which compares the distribution of classifier output, p(y|x), for x from the model to that from test data. It's a pretty different metric from CAFD with different properties, but since you both require a classifier, it would be good to know how the two compare.



Minor points:

In the related work, your discussion of the MMD is misleading: Dziugaite et al. and Li et al. proposed using the MMD for *training* generative models, not for evaluating. Evaluating with two-sample tests based on the MMD using simple kernels was done e.g. by Sutherland et al. (ICLR 2017, https://openreview.net/forum?id=HJWHIKqgl ) and Olmos et al. (https://openreview.net/forum?id=HJWHIKqgl ), and used on top of Inception-like representations e.g. by Lopez-Paz and Oquab (2017), as well as Xu et al. and Binkowski et al. mentioned above.

The derivation (6) of the CAFD is that the derivation (6) is somewhat sloppy about exactly what p() means -- in particular, it's somewhat confusing to use a lowercase p when every distribution you deal with here is actually discrete (for discrete y or for the empirical distribution S, since you're dealing with that and not actually the true distribution of the model, where p(x_i) would not be constant across samples x). It would probably be clearer to distinguish your notation for the true model distribution from the empirical distribution of the S samples.

I don't understand your claim on page 6 that "Unlike Inception Score, because CAFD measures distance on the feature space as FID does, it is able to report overfitting." CAFD, like FID, probably doesn't allow for distributions to appear better than the target in the way that Inception score does. But I don't see how this corresponds to "reporting overfitting"; a model that simply reproduces exactly the empirical distribution of the training set would get an excellent CAFD/FID score, but that's the usual sense of "overfitting."



Overall thoughts:

Using dataset-specific features for evaluation metrics makes a lot of sense, but I don't feel totally satisfied by this paper's investigation of the specific proposal of a VAE, and am particularly worried about whether the metric just ends up preferring models similar to that VAE. I'd really like to see some theoretical and empirical investigation into that.

The CAFD as opposed to FID doesn't feel as nice to me; it's both something of an obvious extension of the previously-used "intra-class FID," and I am also unconvinced by the paper's arguments for its preferability over the FID or other metrics based on image representations like those of Xu et al.

---

### Official Review · AnonReviewer3 · 2018-11-01
**Interesting paper that shows a failure case of FID**

**Rating:** 5
**Confidence:** 3

**Review:**

The paper proposes a new evaluation metric for generative adversarial networks and shows that it is better aligned with human judgment than FID. The metric is based on a domain-specific encoder to extract features of the image rather than ImageNet inception network and a class-aware Frechet distance which makes a Gaussian mixture assumption for the extracted features rather than a simple Gaussian assumption for FID. The paper shows an advantage for the new metric vs the others by constructing examples where FID fails while the proposed metric doesn't. Although this is an interesting finding, it is not a breakthrough in the sense that a domain-specific representation is expected to be better behaved than the features of the inception classifier and using a Gaussian mixture would be an obvious step after FID. Moreover, other metrics don't even rely on any assumption on the features distributions [1,2], so I would expect them to behave at least as well as the proposed metric.


[1] :M. Arjovsky, S. Chintala, L. Bottou, Wasserstein gan
[2] :M. Binkowski, D. J. Sutherland, M. Arbel, and A. Gretton. Demystifying MMD GANs.

---

### Author Response · Authors · 2018-11-26
**Author Response**

Thanks much for your constructive comments and suggestions.

- For the necessity of addressing the sample-based evaluation

FID is a widely used metric for evaluating generative models. However, in our experiments we found that it appeared to be inconsistent with human judgements in some cases. Moreover, we found that there exist potentials to improve the existing FID metric.

- For the domain-specific representations

Our main focus is to address the drawbacks of the ImageNet model on the specific domain e.g. mnist, celebA. We give two seperate proposals including the classifier and the VAE. The experimental results cannot necessarily infer the preferences over the two proposals. Actually, we have not solved the problem of choosing a perfect encoder. We mainly propose to address the necessity of substituting the Imagnet model with a domain-specific encoder to get relatively more meaningful evaluation.

- For the CAFD versus FID
CAFD is indeed an intra-class version over FID. Thus, this is a relatively simple incremental contribution on the previous method.  However, given that FID is widely accepted in the GAN community and used in much literature for evaluation on mnist, fashion-mnist, celebA, etc, we consider it necessary to point it out and conduct both user studies and qualitative experiments to verify the improved effectiveness. Determining the number of modes is a highly non-trivial ill-posed problem, so we choose to use the number of classes for simplicity. MMD and Wasserstein distance are indeed two parallel methods free of the Gaussian assumption. These methods actually make great sense. In this paper, we aim to improve FID and compare our methods only with the baseline. We will delve more into it and maybe more user studies are needed for further comparison.

- Overall

Thanks very much for your kind advice and discussions. We will study more into this serious problem in the future and hope the experimental results in this work can give some inspirations to your concerns. Thank you for your time to review our paper.

---

### Meta-Review · Area_Chair1 · 2018-12-09
**Intersting ideas that need some further investigations**

**Confidence:** 4
**Recommendation:** Reject

**Metareview:**

The paper proposes a novel sample based evaluation metric which extends the idea of FID by replacing the latent features of the inception network by those of a data-set specific (V)AE and the FID by the mean FID of the class-conditional distributions. Furthermore, the paper presents  interesting examples for which FID fails to match the human judgment while the new metric does not. All reviewers agree, that while these ideas are interesting, they are not convinced about the originality and significance of the contribution and believe that the work could be improved by a deeper analysis and experimental investigation.